# The Effect of Changing Fly Ash Content on the Modulus of Compression of Stabilized Soil

**DOI:** 10.3390/ma12182925

**Published:** 2019-09-10

**Authors:** Shuai Yang, Wenbai Liu

**Affiliations:** School of Ocean Science and Engineering, Shanghai Maritime University, Shanghai 201306, China

**Keywords:** curing agent, fly ash, modulus of compression, microscopic parameter

## Abstract

Adding a curing agent can enhance the mechanical properties of soil including its compressive strength. However, few studies have quantitatively analyzed the compressive strength and microstructure of soils to explore the impact of changes in the microstructure on compressive strength. In addition, the cost of curing agents is too high to be widely used. In this study, curing agents with different proportions of fly ash were added to dredger fill to reduce the amount of curing agents needed. The quantitative analysis of the relationships between the modulus of compression *Es* and microstructures of stabilized soil samples is presented. The modulus of compression *Es* was gauged from compression tests. Microscopic images acquired using a scanning electron microscope were processed using the Image-Pro Plus (IPP) image processing software. The microscopic parameters, obtained using IPP, included the average equivalent particle size *Dp*, the average equivalent aperture size *Db*, and the plane pore ratio *e*. This research demonstrated that the fly ash added to the curing agent achieved the same effect as the curing agent, and the amount of curing agent required was reduced. Therefore, the modulus of compression for stabilized soil can be improved. This is due to the hydration products (i.e., calcium silicate hydrate, calcium hydroxide, and ettringite), produced by the hydration reaction, and which adhere to the surface of the particles and fill the spaces among them. Thus, the change in the pore structure and the compactness of the particles helps to increase the modulus of compression. In addition, there was a good linear relationship between the modulus of compression and the microscopic parameters. Using the mathematical relationships between the macroscopic and microscopic parameters, correlations can be built for macro–microscopic research.

## 1. Introduction

With the rapid expansion of China’s coastal economy, a large number of wharf buildings have been constructed, resulting in a large amount of dredger fill. Due to the high-water content of the dredger fill, its engineering properties and performance are very poor. Adding a curing agent to the dredger fill can quickly increase the quality of the performance; however, curing agents are not widely used due to the fact of their high cost [1,2,3,4,5,6,7,8,9].

Fly ash was once regarded as waste, which has created serious environmental stressors in China. If fly ash, possessing the properties of volcanic ash, can effectively improve the performance of curing agents, the cost of acquiring these curing agents and the release of pollutants due to the use fly ash will be reduced, which is of great importance to waste utilization as a waste management system. Therefore, we investigated whether fly ash can improve the properties of soil in order to increase the amount of fly ash and decrease the amount of curing agent used [10,11,12,13,14,15,16].

In engineering applications, the modulus of the compression of soil is an important property. However, current research has not yet identified a good correlation between the macroscopic and microscopic properties of soil, and there are relatively few quantitative studies on these properties. Reference [1] studied the effects of pumice-based porous material on the hydration characteristics and persistent shrinkage of ultra-high-performance concrete. In addition, Reference [2] conducted a study on the feasibility of using agricultural biomass fly ash for soil stabilization in road works. Reference [4] presented an investigation into the development and characterization of glass–ceramics from combinations of slag, fly ash, and glass cullet without adding nucleating agents. Furthermore, Reference [5] studied the microstructural and mechanical properties of WC-10Ni3Al-cemented carbide prepared using different ball-milling suspensions. Though the mechanical properties including the modulus of compression of the stabilized soils can be improved using various curing agents, fly ash is seldom used in stable soils due to the lack of economic benefits. In this study, a curing agent mixed with fly ash was added to a stabilized soil to improve the modulus of compression. The use of fly ash in stable soil reduces environmental pressures [17]. Hence, a significant reduction in environmental problems can be achieved by increasing the utilization of fly ash in stabilized soil [18]. In addition, there is little quantitative research on the macroscopic and microscopic properties of fly ash. Only a rough trend in the microstructural parameters and modulus of compression have been discussed with changing fly ash content and curing period [17]. However, this research demonstrated the numerical relationship between the modulus of compression and microscopic parameters. When the modulus of compression was measured using the compression test, the microscopic parameters were calculated using the mathematical relationships. Therefore, the use of microscopic experiments and software processing can be omitted when obtaining the microscopic parameters.

In this paper, by increasing the amount of fly ash in samples of stable soil, the proportion of curing agent needed was reduced. The effects of curing time and fly ash contents on stabilized soils were investigated. The modulus of compression and microstructural parameters were obtained using compression tests and SEM. Based on this, the quantitative correlations, not just the trends, of the macroscopic and microscopic parameters of fly ash in stable soil were established, revealing how the microscopic parameters changed as the macroscopic parameters changed as well.

## 2. Materials and Methods

### 2.1. Sample Preparation

The soil used in this study was from Hengsha Island close to Shanghai. Because the water content of the mud–water mixture in the dredging pipeline was high, a mixture with 40% water content was selected to simulate the drainage filling conditions of the dredged fill. The specimens were separated into four groups and each sample had a moisture content of 40%, a curing agent content of 3%, and fly ash contents of 0%, 2%, 4%, and 8%, respectively. In addition, to determine the influence of fly ash on the curing agent, a sample with 4% curing agent content and 0% fly ash content was added to the group. The mold used to prepare the soil samples was a cutting ring, which was 19.9 mm in height and 61.7 mm in diameter. To determine the quantitative relationship between the macroscopic and microscopic properties, the curing ages for the samples were 7 days, 14 days, 21 days, 28 days, 35 days, 42 days, 56 days, 72 days, 90 days, and 180 days. This timeline was chosen because significant sample changes occur during the early periods of curing, while fewer changes occur later in the curing process. Therefore, the intervals between the early periods of curing were short, but the ones during the later periods were longer.

Before the preparation of soil samples, the soil was dried to remove a few large particle impurities in the undisturbed sand. The physical properties of the soil are reported in Table 1. The specific gravity, unit weight, and water content of the soil, carried out in accordance with the Standard SL 237-1999, were respectively 2.69, 16.01 kN/m^3^, and 24%. The liquid limit and plastic limit were, respectively, measured as *w_L_* = 37% and *w_P_* = 26% in accordance with the Standard GB 50007-2002. The Internal friction angle and the cohesion of the soil, measured by the direct shear test, were 7° and 0.826 kPa in accordance with Standard GB/T 50123-1999.

During sample preparation, the curing agent and fly ash were added to the dry undisturbed soil following Standard GB 50010-2010. Then, water was added (until a water content of 40% was reached) and the soil was mixed well. In addition, the mixed soil sample was placed in a mold two to three times. Each time it was tampered down with a tamping rod. After this, the entire soil sample and the mold were placed in the curing box at 20 °C and a humidity of 99% for the curing of the soil. Finally, the sample was removed from the curing box to continue the follow-up experiment.

The curing agent content, the fly ash content, and the water content were, respectively, defined as:(1)$$w_{c}=\frac{m_{c}}{m_{s}}\times 100\%$$
(2)$$w_{FA}=\frac{m_{FA}}{m_{s}}\times 100\%$$
(3)$$w_{w}=\frac{m_{w}}{m_{s}}\times 100\%$$
where *w_c_* = the curing agent content by weight of dry soil (%); *w_FA_* = the fly ash content by weight of dry soil (%); *w_w_* = the water content by weight of dry soil (%); *m_s_* = mass of the dry undisturbed soil (kg); *m_c_* = mass of the curing agent (kg); *m_FA_* = mass of the fly ash (kg); *m_w_* = mass of the water (kg).

The curing agent used in the test was a self-made curing agent, which contained a variety of materials, including ordinary Portland cement (OPC), lime (CaO), sodium sulfonate, acrylamide, etc. We evaluated the chemical composition of the curing agent and fly ash by X-ray fluorescence (XRF, PANalytical B.V., Beijing, China) [18]. The main components of the curing agent were CaO and SiO_2_. The main components of the fly ash and curing agent are reported in Table 2 and Table 3.

### 2.2. Test Method

The entire experimental process was as follows. First, fly ash (0%, 2%, 4%, and 8%) and curing agent (3%) were added to the soil to prepare the samples. In addition, to determine the influence of the fly ash on the curing agent, fly ash (0%) and the curing agent (4%) were added to soil to prepare another sample. Then, the samples were placed in the curing box. In addition, the compression test was conducted to determine the modulus of compression of the samples. Meanwhile, microscopic images of the samples (fly ash contents of 0%, 2%, 4%, and 8%, curing agent content of 3%) were obtained using a scanning electron microscope (SEM). Next, the microscopic images were processed using the Image-Pro Plus (IPP) image processing software, and the microstructural parameters were obtained. Based on the above described macroscopic and microscopic data, the corresponding microstructure and modulus of compression were, respectively, taken as the *x*-axis and the *y*-axis for a certain curing period and one amount of fly ash. Then, a rectangular coordinate system was established and a fitting curve was drawn. Finally, the quantitative dependence of the modulus of compression and the microstructural parameters was formed.

#### 2.2.1. The Compression Test

The process of the compression test followed the Standard GB T50123-1999. The soil samples taken from the curing box were immersed in water for 1 day before the compression test. Different vertical loads (p) were imposed on the soil samples according to 50 kPa, 100 kPa, and 200 kPa, i.e., the load rate was 1, 0.5, and 0.5. When the vertical deformation of the specimen was basically stable after each vertical load was imposed, the vertical deformation *h*_1_, *h*_2,_ and *h*_3_ were recorded. Therefore, the variation of the vertical deformation of each vertical load was ∆*h*_1_, ∆*h*_2_, and ∆*h*_3_. Then, the void ratio $e_{i}$ under each vertical load was calculated using the following formula. Further details of the compression test are described in References [19,20].
(4)$$e_{i}=e_{0}-\frac{1+e_{0}}{h_{0}}\Delta h_{i}$$
where $e_{i}$ is the void ratio after consolidation and stabilization of the specimens under different pressure levels; $e_{0}$ is the initial void ratio without loading; and $\Delta h_{i}$ is the variation of vertical deformation under different loads.

Finally, the coefficient of compressibility and the modulus of compression were determined. The coefficient of compressibility for a certain pressure range in accordance with Standard GB T50123-1999 is as follow: (5)$$a_{v}=\frac{e_{i}-e_{i+1}}{p_{i+1}-p_{i}}$$
where $a_{v}$ is the coefficient of compressibility (MPa^−1^); $p_{i}$ is the pressure value at a certain level (MPa); and $e_{i}$ is the void ratio at a certain level.

The modulus of compression of the samples is: (6)$$E_{s}=\frac{1+e_{0}}{a_{v}}$$
where $E_{s}$ is the compression modulus (MPa); $\;e_{0}$ is the initial void ratio without loading; and $a_{v}$ is the coefficient of compressibility (MPa^−1^).

#### 2.2.2. SEM Test

The SEM analysis was conducted using a field emission scanning electron microscope (ER-SEM) made in Tokyo, Japan (JSM7610F). The details of the test process are described in References [16,21]. The following SEM analytical procedure was used.
Soil samples were removed from the curing box. Then, the soil samples were removed from the mold and cut into small 10 × 10 × 20 mm samples with a knife.The samples were dehydrated for one day before the scanning electron microscopy analysis. The oven drying method was adopted to dehydrate the samples. The temperature was 110 °C.After dehydration and drying, the specimens were broken and the SEM analysis was performed on the flat fracture surface. If the surface of a sample was not flat enough, the brightness of the SEM image deviated and the resulting image was not clear enough.Because the samples were not conductive, it was necessary to spray gold on the samples. When spraying the gold, we were careful to control the spraying time. Spraying for a too long time leads to an excessively thick gold film on a sample’s surface, which may cover the undisturbed structure of the stabilized soil. Spraying for a too short time results in insufficient conductivity and unclear images. In this test, the sputtering speed was 10 mm/min and the sputtering time was 60 s.Then, scanning electron microscopy was conducted.

#### 2.2.3. IPP Image Processing

The microscopic images of the stabilized soil were processed and analyzed using IPP. The processing steps were as follows. More details of specific practices about IPP are described in References [3,15,16].
Image segmentation is defined as the binarization of an image, and the images are converted into the pictures which best reflects the true appearance of the stabilized soil in the IPP software.Image morphology processing was conducted in order to avoid destroying the original image structure.The measuring units were calibrated. The image is in pixels, but our research requires an area size, so we needed to add new rulers using the software. It can be calibrated using the scale of the SEM image.The measurement parameters for the measurement were selected. The number and area of the pores and particles needed to be extracted in this study.The measurement data were extracted. After the measurement was completed, the measurement data, including the number and area of the pores and the particles, were extracted and analyzed.

## 3. Test Results

### 3.1. Compression Test and SEM Results

The modulus of compression values of the stabilized soils for different curing periods and curing agents (3% and 4%) with different fly ash contents are reported in Table 4. As the curing period increased, the modulus of compression of the stabilized soil increased. 

The relationship between the modulus of compression and the curing period is reported in Figure 1. Figure 1 illustrates that the modulus of compression of the solidified calcareous sand increased with the increase of curing time, and the modulus of compression did not increase after 90 days of curing. The modulus of compression for curve 2 was higher than that of curve 5. This demonstrates that fly ash added to the curing agent can play the role of a curing agent, and it can replace part of the curing agent. It also indicates that after increasing the fly ash content, the amount of curing agent needed decreased.

This was due to the hydration reaction of the curing agent and the fly ash. Because the curing agent mainly contains CaO and SiO_2_, the hydration reaction of the curing agent is as follows [17,22,23,24]:$$3\mathrm{CaO}\cdot {\mathrm{SiO}}_{2}+7H_{2}O\;\to 3\mathrm{CaO}\cdot 2{\mathrm{SiO}}_{2}\cdot 3H_{2}O+3\mathrm{Ca}{\left(\mathrm{OH}\right)}_{2}$$

The calcium silicate is hydrated by the hydration reaction [21]. Its microstructure is flocculent, and it possesses strong cementation [22]. The calcium hydroxide, which has a flaky microstructure, adheres to the particle surfaces or fills in the pores, so it plays a role in stabilizing the soil.

In addition, the fly ash possesses the pozzolanic activity [18]. As reported in Reference [17], the fly ash reacted and contained the active aluminum (Al_2_O_3_). In the alkaline environment of Ca(OH)_2_, the active substance (Al_2_O_3_) in the fly ash reacted [23]. Therefore, the fly ash and calcium hydroxide undergo a secondary hydration reaction. The hydration reaction is as follows [17]:$${\mathrm{Al}}_{2}O_{3}+{\;\mathrm{CaSO}}_{4}\cdot 2H_{2}O+\;3\mathrm{Ca}{\left(\mathrm{OH}\right)}_{2}+7H_{2}O\;\to 4\mathrm{CaO}\cdot {\mathrm{Al}}_{2}O_{3}\cdot {\mathrm{SO}}_{3}\cdot 12H_{2}O$$

The ettringite produced causes volume expansion, which exert an extrusion effect on the particles and make the soil more compact [17]. Therefore, the ettringite contributes to the increase in the modulus of compression of the stabilized soil.

The secondary hydration reaction of the fly ash consumed the calcium hydroxide produced by the hydration reaction of the curing agent. This helped to increase the hydration reaction of the curing agent to produce more calcium hydroxide. In addition, the ettringite, which is beneficial to the modulus of compression of the stabilized soil, was produced by the hydration reaction of the fly ash. Therefore, the fly ash added in the curing agent achieved the effect of curing agent by promoting the hydration reaction of the curing agent and producing the hydration product (i.e., ettringite). That is to say, the mixture obtained by adding fly ash to the curing agent can act as a curing agent, and the amount of curing agent is then reduced. This demonstrates the reason why the modulus of compression for curve 2 was higher than that of curve 5 in Figure 1.

Based on the above analysis, the curing agent mixed with fly ash improved the modulus of compression of the stabilized soil due to the hydration products produced by the hydration reaction. According to References [22,24], the hydration products include flocculent hydrated calcium silicate (Figure 2a), flaky calcium hydroxide (Figure 2b), and acicular ettringite (Figure 2c).

The microscopic appearance of the stabilized soil is shown in Figure 3. According to Figure 3, the surface of the particles adhered to many substances and becomes rougher. This was because the hydration products (i.e., calcium silicate hydrate and calcium hydroxide) and ettringite adhered to the surface of the particles and filled the voids among the particles [18]. Thus, the modulus of compression of the stabilized soil was improved.

### 3.2. Image Analysis Results

The SEM images of the stabilized soils were processed using IPP, and the data related to the particles and pores were extracted from the image analysis. The average equivalent particle size *D_p_*, the average equivalent aperture size *D_b_*, and the plane pore ratio *e* were quantitatively analyzed.

The average equivalent particle size *D_p_* is the average value of the equivalent diameter of all of the particles in the analyzed area. 

The calculation formula is as follows:(7)$$D_{p}=\sum d_{i}/n\;$$
(8)$$d_{i}=2\pi^{\frac{1}{2}}A_{i}^{\frac{1}{2}}$$
where $d_{i}$ is the diameter of the equivalent circle equal to the area of the particle unit (μm); $A_{i}$ is the area of the particle unit (μm^2^); and n is the number of particles in the analyzed area.

Table 4 reports the average equivalent particle size for different curing periods and fly ash contents. According to Table 5, as the fly ash content increased, the average equivalent particle sizes increased.

The average equivalent aperture size *D_b_* is the average value of the equivalent diameter of all of the pores in the analyzed area. 

The calculation formula is as follows:(9)$$D_{b}=\sum d_{i}/N$$
(10)$$d_{i}=2\pi^{\frac{1}{2}}A_{i}^{\frac{1}{2}}\;$$
where $d_{i}$ is the diameter of the equivalent circle that is equal to the area of the pore unit (μm); $A_{i}$ is the area of the pore unit (μm^2^); and N is the number of pores in the analyzed area.

The average equivalent aperture size of the stabilized soils for different curing periods and fly ash contents are reported in Table 6. As the content of fly ash increased, the average equivalent aperture sizes decreased.

The plane pore ratio *e* is the ratio of the total area of the pores to the total area of the particles in the analyzed area. 

The calculation formula is as follows:(11)$$e=S_{b}/S_{p}$$
where e is the plane pore ratio; $S_{p}$ is the particle area in the analyzed area (μm^2^); and $S_{b}$ is the pore area in the analyzed area (μm^2^).

The plane pore ratio of the stabilized soils for different curing periods and fly ash contents are reported in Table 7. With the increase in fly ash content, the plane pore ratios decreased.

## 4. Relationships Among the Parameters and a Discussion

The modulus of compression of stabilized soils for different curing periods and fly ash contents can be obtained based on the results of the compression tests. Furthermore, the microscopic images of the stabilized soil samples (fly ash contents of 0%, 2%, 4%, and 8%, curing agent content of 3%) were obtained from the SEM analysis. After the images were processed using IPP and then computed, the microscopic parameters of the stabilized soil were obtained for different curing periods and fly ash contents. Based on the above macroscopic and microscopic data, for a certain curing period and fly ash content, the microscopic parameters and modulus of compression were obtained, and then they were used as the *x*-axis and *y*-axis, respectively. Image coordinates were established and the fitting curves among these parameters were drawn. Using this process, the quantitative dependencies between the modulus of compression and the microscopic parameters of the soil samples were obtained.

The relationships between the modulus of compression and the average equivalent particle size of the soil samples are shown in Figure 4. According to Figure 4, when the content of fly ash remained the same and the average equivalent particle size increased, the modulus of compression rose linearly. During the solidification process of the soil samples, the microcosmic aspect was the linear increase in the average equivalent particle size, while the macroscopic aspect was the linear increase in the compression modulus. In addition, due to the hydration reaction of the curing agent and the fly ash, the hydration products (i.e., calcium silicate hydrate, calcium hydroxide, and ettringite) were attached to the surface of the particles, resulting in an increase in the average equivalent particle size. Therefore, the increased compactness of the particles contributed to the modulus of compression.

Table 8 illustrates the linear dependencies of the modulus of compression and the average equivalent particle sizes of the soil samples. The modulus of compression and the average equivalent particle sizes were linearly related for fly ash contents of 0%, 2%, 4%, and 8%. Consequently, the modulus of compression can be defined for a given fly ash content and curing period and, then, the average equivalent particle size can be calculated using the above linear formula.

Figure 5 shows the relationships between the modulus of compression and the average equivalent aperture sizes of the soil samples. As the average equivalent aperture sizes decreased, the modulus of compression increased linearly. This indicates that for the same fly ash content, the microcosmic aspect was the decrease in the average equivalent aperture size, while the macroscopic aspect was the linear increase in the modulus of compression. In addition, according to the analysis of the previous SEM images, the hydration products filled the pores among the particles. Therefore, the hydration products made the particles more compact, enhancing the modulus of compression.

The linear relationships between the modulus of compression and the average equivalent aperture sizes of the stabilized soil samples are reported in Table 9. Table 9 also shows the linear formulas describing the relationships between the modulus of compression and the average equivalent aperture sizes for fly ash contents of 0%, 2%, 4%, and 8%. Similarly, the average equivalent aperture sizes of the stabilized soil samples can be calculated using these linear relationships.

The dependencies between the modulus of compression and the plane pore ratios are shown in Figure 6. Figure 6 illustrates that when the fly ash content remained the same and as the plane pore ratio decreased, the modulus of compression increased linearly. In other words, the microcosmic aspect was the decrease in the plane pore ratio, while the macroscopic aspect was the linear increase in the modulus of compression.

The linear relationships between the modulus of compression and the plane pore ratios of the stabilized soil samples are shown in Table 10. The table also reveals that there were linear relationships between the modulus of compression and the plane pore ratios for fly ash contents of 0%, 2%, 4%, and 8%. Similarly, the plane pore ratio can be calculated using these formulas. 

Figure 4, Figure 5 and Figure 6 show that with a larger average equivalent particle size *D_p_*, a smaller average equivalent aperture size *D_b_*, and a smaller plane pore ratio *e*, as the curing time increased, the modulus of compression increased. These results, combined with the microscopic analysis of the previous SEM images, indicate that the increase in the modulus of compression may be caused by the hydration products (i.e., calcium silicate hydrate, calcium hydroxide, and ettringite), which fill the pores among the particles and changes the pore structure.

In this paper, we showed that fly ash added to the curing agent can achieve the effect of a curing agent by promoting the hydration reaction of the curing agent and producing the hydration products. That is to say, when fly ash is added to the curing agent, it can replace part of the curing agent [22], and the amount of curing agent needed is reduced. Therefore, the addition of fly ash can improve the modulus of compression of stabilized soil, as shown in Figure 1. According to References [22,23], this is because the hydration reaction of the curing agent produces the hydration products, i.e., calcium silicate hydrate and calcium hydroxide. In addition, fly ash possesses pozzolanic activity, and it contains active aluminum (Al_2_O_3_) [18,25]. As reported in References [17], ettringite is produced by the hydration reaction of fly ash. These hydration products (i.e., calcium silicate hydrate, calcium hydroxide, and ettringite), adhere to the surface of the particles and fill the spaces among the particles [24], which has a significant effect on the compression modulus of the stabilized soil [26]. The addition of fly ash in the curing agent contributes to the utilization of fly ash and plays a role in waste utilization. In addition, Figure 1 indicates that after increasing the fly ash content, the amount of curing agent needed decreased. The use of fly ash has implications for environmental protection and cost reduction [18,27,28].

The modulus of compression of the stabilized soil samples had a good linear correlation with the three microstructural parameters. Based on this study, we concluded that the modulus of compression can be determined based on the fly ash content (2%, 4%, and 8%) and curing period (7 d, 14 d, 21 d, 28 d, 35 d, 42 d, 56 d, 72 d, 90 d, and 180 d). Therefore, the microstructural parameters can be calculated using the macroscopic–microscopic relationships obtained from the experiment. According to Reference [29], the compressive strength and microstructure of alkali-activated fly ash/slag binders at high temperatures were proposed. Reference [30] studied the compressive strength and microstructure of alkali-activated blast furnace slag/sewage sludge ash (GGBS/SSA) blends cured at room temperature. Furthermore, Reference [31] studied the comparative performance of alkali activated slag/metakaolin cement pastes exposed to high temperatures. Based on the above studies, there is little quantitative research on the macroscopic and microscopic properties of fly ash. Only the changing trends between the macroscopic and microscopic parameters were determined and these studies lacked sufficient correlation. However, this study demonstrated the numerical relationship between the modulus of compression and microscopic parameters. Therefore, when the modulus of compression was measured using the compression test, the microscopic parameters were obtained using a mathematical formula. In this way, the use of microscopic experiments and software processing were avoided.

Though there are significant findings demonstrated in this the paper, it still has limitations. For instance, the amount of fly ash utilized was only 2%, 4%, and 8%, which needs to be further expanded. In addition, one subject that remains to be explored is how to further reduce the amount of curing agent used via an increase in the amount of fly ash used and then, how to practically apply these finding in real-world engineering construction projects.. 

## 5. Conclusions

In this study, the moduli of compression, *E_s_*, for stabilized soil samples were obtained using compression tests. The microscopic SEM images were analyzed using IPP. The quantitative analysis of the relationships between the modulus of compression and the microstructural parameters were determined. Based on our results, we reached the following conclusions.
Fly ash added to the curing agent can achieve the same effect as a curing agent, and the amount of curing agent required is reduced. The hydration products (i.e., calcium silicate hydrate, calcium hydroxide, and ettringite) produced by the hydration reaction of the curing agent and the fly ash adhere to the surface of the particles and fill the spaces among the particles. Therefore, the change in the pore structure and the compactness of the particles contributes to the increase in the modulus of compression.The modulus of compression has a strong linear correlation with the three microstructural parameters. By establishing the quantitative relationship between the macroscopic and microscopic parameters, the modulus of compression can be defined based on the fly ash content (2%, 4%, and 8%) and the curing period (7 d, 14 d, 21 d, 28 d, 35 d, 42 d, 56 d, 72 d, 90 d, and 180 d). Using the mathematical relationships between the macroscopic and microscopic parameters, the microstructural parameters can be calculated and the correlations can be built for macro–microscopic research. In this way, the use of complex tests (i.e., SEM) and software operation processes (i.e., IPP) can be avoided when obtaining the microscopic parameters.

## Figures and Tables

**Figure 1 materials-12-02925-f001:**
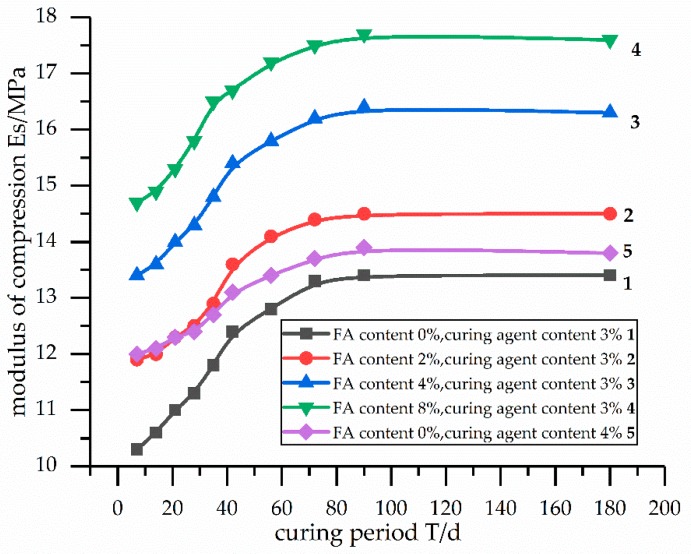
Relationship between the modulus of compression and the curing period.

**Figure 2 materials-12-02925-f002:**
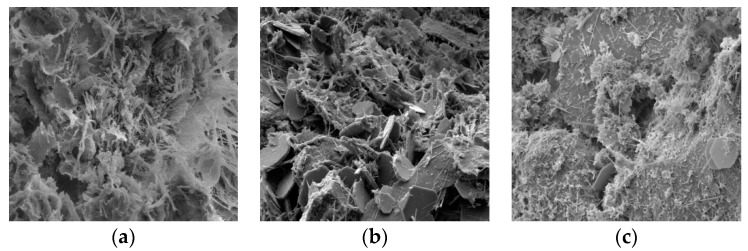
Microscopic appearance of the calcium silicate hydrate, calcium hydroxide, and ettringite. (**a**) calcium silicate hydrate; (**b**) calcium hydroxide; (**c**) ettringite.

**Figure 3 materials-12-02925-f003:**
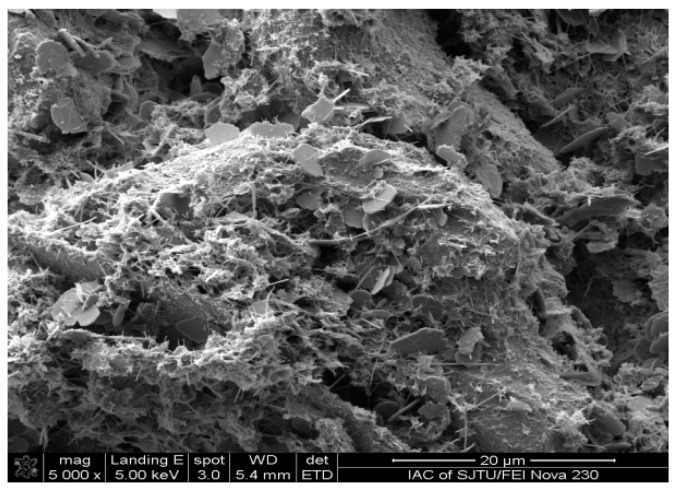
Microscopic appearance of the stabilized soil.

**Figure 4 materials-12-02925-f004:**
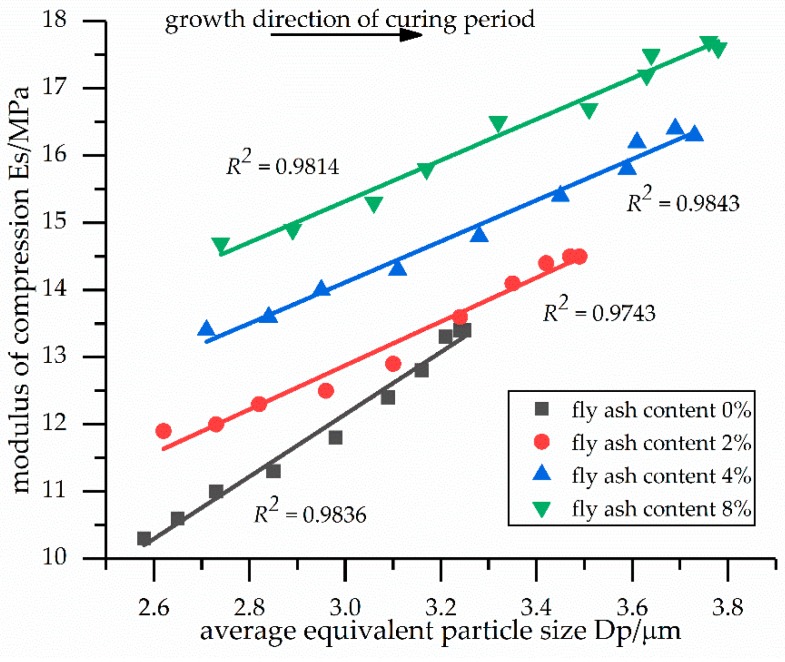
Relationships between the modulus of compression and the average equivalent particle sizes of the soil samples.

**Figure 5 materials-12-02925-f005:**
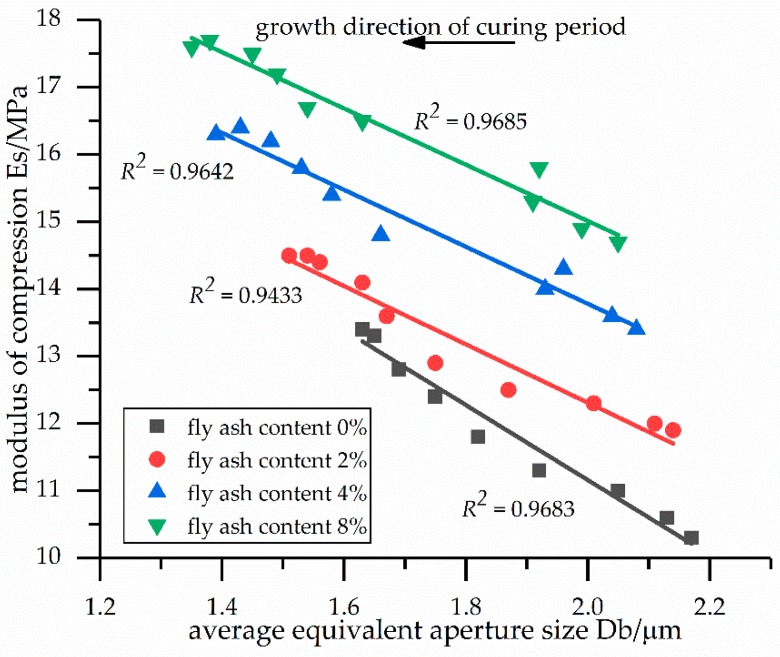
Relationships between the modulus of compression and the average equivalent aperture sizes of the soil samples.

**Figure 6 materials-12-02925-f006:**
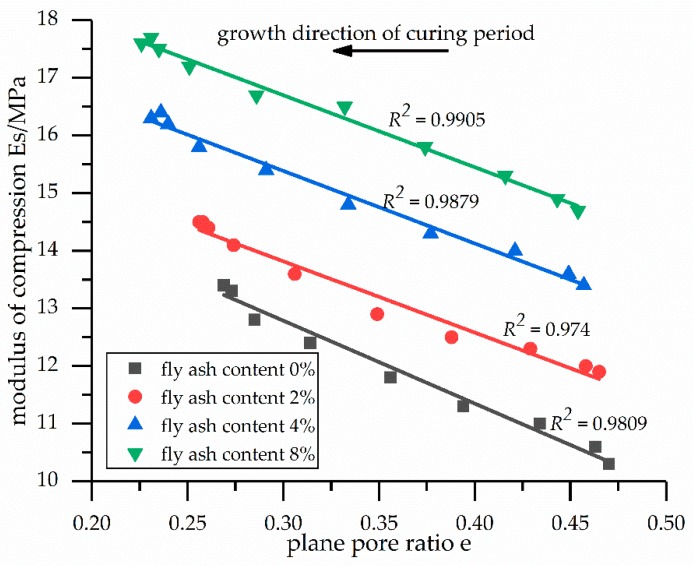
Dependencies between the modulus of compression and the plane pore ratios of the soil samples.

**Table 1 materials-12-02925-t001:** Physical properties of the soil.

Specific Gravity	Unit Weight (kN/m^3^)	Water Content (%)	Liquid Limit	Plastic Limit	Internal Friction Angle (°)	Cohesion (kPa)
2.69	16.01	24	37	26	7	0.826

**Table 2 materials-12-02925-t002:** Main components of the fly ash.

SiO_2_	Al_2_O_3_	Fe_2_O_3_	CaO	MgO	SO_3_	K_2_O	Na_2_O
40.7	28.9	6.42	1.93	1.73	0.45	0.59	0.48

**Table 3 materials-12-02925-t003:** Main components of the curing agent (%).

SiO_2_	Al_2_O_3_	Fe_2_O_3_	CaO	MgO	K_2_O	Na_2_O	P_2_O_3_	TiO_2_	SO_3_
21.04	6.94	2.36	61.27	1.32	1.02	0.27	0.12	0.38	2.31

**Table 4 materials-12-02925-t004:** The modulus of compression of the stabilized soil (MPa).

Curing Period T (d)	Curing Agent Content (%)				
	3				4
	Content of Fly Ash λ (%)				
	0	2	4	8	0
7	10.3	11.9	13.4	14.7	12.0
14	10.6	12	13.6	14.9	12.1
21	11	12.3	14	15.3	12.3
28	11.3	12.5	14.3	15.8	12.4
35	11.8	12.9	14.8	16.5	12.7
42	12.4	13.6	15.4	16.7	13.1
56	12.8	14.1	15.8	17.2	13.4
72	13.3	14.4	16.2	17.5	13.7
90	13.4	14.5	16.4	17.7	13.9
180	13.4	14.5	16.3	17.6	13.8

**Table 5 materials-12-02925-t005:** The average equivalent particle sizes of the stabilized soils (μm).

Curing Period T (d)	Content of Fly Ash λ (%)			
	0	2	4	8
7	2.58	2.62	2.71	2.74
14	2.65	2.73	2.84	2.89
21	2.73	2.82	2.95	3.06
28	2.85	2.96	3.11	3.17
35	2.98	3.10	3.28	3.32
42	3.09	3.24	3.45	3.51
56	3.16	3.35	3.59	3.63
72	3.21	3.42	3.61	3.64
90	3.24	3.47	3.69	3.76
180	3.25	3.49	3.73	3.78

**Table 6 materials-12-02925-t006:** The average equivalent aperture sizes of the stabilized soils (μm).

Curing Period T (d)	Content of Fly Ash λ (%)			
	0	2	4	8
7	2.17	2.14	2.08	2.05
14	2.13	2.11	2.04	1.99
21	2.05	2.01	1.93	1.91
28	1.92	1.87	1.96	1.92
35	1.82	1.75	1.66	1.63
42	1.75	1.67	1.58	1.54
56	1.69	1.63	1.53	1.49
72	1.65	1.56	1.48	1.45
90	1.63	1.54	1.43	1.38
180	1.63	1.51	1.39	1.35

**Table 7 materials-12-02925-t007:** The plane pore ratios of the stabilized soils.

Curing Period T (d)	Content of Fly Ash λ (%)			
	0	2	4	8
7	0.47	0.465	0.457	0.454
14	0.463	0.458	0.449	0.443
21	0.434	0.429	0.421	0.416
28	0.394	0.388	0.377	0.374
35	0.356	0.349	0.334	0.332
42	0.314	0.306	0.291	0.286
56	0.285	0.274	0.256	0.251
72	0.273	0.261	0.24	0.235
90	0.269	0.258	0.236	0.231
180	0.269	0.256	0.231	0.226

**Table 8 materials-12-02925-t008:** Linear dependencies of the modulus of compression and the average equivalent particle sizes of the stabilized soils.

Average Equivalent Particle Size, *D_p_* (μm)			
Fly Ash Content λ (%)	Linear Relationships between the Modulus of Compression and the Average Equivalent Particle Sizes	*R* ^2^	Slope K
0	*Es* = 4.6323 *Dp* − 1.7465	0.9836	4.6323
2	*Es* = 3.2702 *Dp* + 3.0669	0.9743	3.2702
4	*Es* = 3.0557 *Dp* + 4.9484	0.9843	3.0557
8	*Es* = 3.0518 *Dp* + 6.1663	0.9814	3.0518

**Table 9 materials-12-02925-t009:** Linear relationships between the modulus of compression and the average equivalent aperture sizes of the stabilized soil samples.

Average Equivalent Aperture Size, *D_b_* (μm)			
Fly Ash Content λ (%)	Linear Relationships between the Modulus of Compression and the Average Equivalent Aperture Sizes	*R* ^2^	Slope K
0	*Es* = −5.5719 *Db* + 22.305	0.9683	5.5719
2	*Es* = −4.34 *Db* + 20.991	0.9433	4.34
4	*Es* = −4.2335 *Db* + 22.251	0.9642	4.2335
8	*Es* = −4.1825 *Db* + 23.379	0.9685	4.1825

**Table 10 materials-12-02925-t010:** Linear relationships between the modulus of compression and the plane pore ratios of the stabilized soil samples.

Plane Pore Ratio (*e*)			
Fly Ash Content λ (%)	Linear Relationships between the Modulus of Compression and the Plane Pore Ratios	*R* ^2^	Slope K
0	*Es* = −14.372 *e* + 17.099	0.9809	14.372
2	*Es* = −12.381 *e* + 17.534	0.974	12.381
4	*Es* = −12.579 *e* + 19.161	0.9879	12.579
8	*Es* = −12.475 *e* + 20.442	0.9905	12.475

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
