# Peer review of "The Effect of Changing Fly Ash Content on the Modulus of Compression of Stabilized Soil"

_materials, 2019, doi:10.3390/ma12182925_

Round 1
Reviewer 1 Report
Please find attached a PDF file with my comments and suggestions for authors.

Reviewer 2 Report
The paper talks about changes in compressive modulus of a natural soil sample treated with a curing agent and varied amounts of fly ash. It draws correlations between the compressive modulus (CM) and soil properties obtained from SEM imaging (e.g. particles and voids sizes).
Based on the study the paper comes up with correlations between microscopic properties and CM that allow predicting the modulus at different curing ages of the soil with a given fly ash, curing agent contents using processed SEM images.
There are several issues with the work that should be addressed throughout the manuscript.
Firstly, the Introduction does not give enough information to understand what this paper is building upon. The general statements like “All these studies have limitations” should be removed or specifics should be given. The discussion of the previous work limitations vs. what is done in the present one is confusingly spread between the two sections “introduction” and “discussion”, it could be easier to understand the previous studies and their limitations compared against the current one discussed in the same secion. The introduction could also benefit from an explanation why quantitative correlation of micro- and macro- parameters is beneficial – it is not obvious from the paper whether it is easier to process SEM images and then calculate the CM based on the provided quantitative relations or to determine the CM directly by testing.
Secondly, “materials and methods” part misses some key information (e.g. what curing agent was used? What given percentages refer to – weight or volume? By weight/volume of what?). On the other hand, some information is repeatedly given (see details below). As a result, it is very difficult to understand some of the statements as, for example, decreased amount of curing agent with increased amount of fly ash, while it is stated that the curing agent content was 3% in all blends. Because there is no information on the chemistry of the curing agent it is impossible to know whether the proposed reactions are relevant and involve only the curing agent or fly ash as well.
In the same section the information given for the SEM analyses is not clear – it is either too detailed if it follows the procedure given in the reference or not sufficient if the procedure is proposed by the authors themselves (it should be explained what a reader should look for in the figures).
In the “test results” “discussion” and “conclusions” sections it should be clarified how based on the presented experimental information the authors made a number of the statements and the proposed conclusions (see details below). For example, without any experimental evidence it is stated repeatedly that fly ash reacts (“hydrates”) enhancing efficiency of the curing agent.
Detailed comments:
Lines 45-46 and further on – the references should give the last and not the first names of the authors, this needs to be corrected both throughout the text and in the “references” section.
Line 50 – general statements without supporting information should be avoided (see above).
Line 57 – “dependence” of macro- and micro- parameters probably refers to “correlation” between them.
Line 59 – “reasonably and intuitively” does not really belong here, the work makes quantitative predictions on changes in macroscopic parameters with microscopic ones.
Lines 55 and 66 – the composition of the curing agent should be given otherwise the mechanisms of increasing CM of the soil given further are arbitrary.
The percentage should be defined – mass or volume and percentage of total blend, soil or something else?
Lines 78- 84 – this information repeats the information given in lines 65-70.
Line 79 – does 40% of water account for the water content of the soil or not?
Table 2 – what methods were used to determine the components of fly ash?
Line 93 – “meanwhile”? – were the images taken after the curing?
Lines 107-116 – what is the reason to give these equations? If there is a good reason to give them all the parameters in the equations should be defined.
Line 118 – “SEM made in Japan” – microscope model should be given.
Lines 118-119 – “Yang et al. [16] and Natnael et al. [17] have introduced SEM” – the SEM was clearly introduced earlier than that. Do you mean for this type of studies?
Line 123 – what temperature?
Lines 133-134 – this point repeats the one above, should be combined.
Lines 135-157 – this is not clear – what a reader is supposed to see in the given figures? What do “the measurement data” refer to? What kind of “image processing” was used? What thresholds the authors refer to?
Lines 165-166 – why there are 2 reactions given? The second reaction is the first one divided by 1.5.
Line 170 – this is not a hydration reaction; hydration reactions involve water.
The reactions given in lines 165-171 are impossible to interpret without knowing the composition of the curing agent. The given products would form If the curing agent was OPC without any involvement of fly ash. There is no experimental information that would allow to conclude that fly ash reacted with the curing agent.
Lines 174-175 there are no experimental evidence in the paper that fly ash reacts.
Figure 4 – how were these products identified? – either a reference or elemental composition should be given.
Lines 197-183 – there is no evidence in Figure 5 that fly ash reacted. It is not clear how the conclusion on fly ash reactions was made. Reaction with portlandite is not a hydration reaction.
Line 206 – should be “decreased”
Line 222 – “corresponding”? correlation between CM and microscopic parameters?
Tables 7-9 repeat information given in Figures, they are not needed.
Lines 260-262 – what experimental observations do show this? Is it just a common-sense statement?
Line 294 - There were no experimental results showing that fly ash improved the efficiency of the curing agent. The results showed that in the presence of fly ash the pore size decreased and the particle size increased. This does not prove that there were reactions between fly ash and curing agent. Could that be a result of improved packing in the presence of fly ash?
Lines 295-296 – Are these hydration products of the curing agent?
Lines 300-301 - This is not clear: if the curing agent was added at 3% by weight of blend its amount did not decrease.
Line 302 - Not clear what was done in the reference that was improved upon by the current study.
Lines 306-310 - This is not clear; the discussion of literature results should be improved in the introduction. This does not belong in the discussion of the paper results.
Lines 313-314 – the amount of the curing agent was 3% - how was it reduced?
The discussion should address a possible packing effect of added fly ash that would increase solid volume fraction and decrease the voids without any chemical reactions of fly ash.
Line 320 – what specific results do show that fly ash enhances effect of curing agent?
Lines 320-321 – are these hydration products of the curing agent or reaction products of fly ash?
Lines 330-331 – what is the advantage of calculating vs. measuring the CM? Is not it easier to measure CM than to process the images?
Reviewer 3 Report
The authors need to pay attention to the following comments:
Page 1: “compression modulus” is not a common terminology. Please review and revise.Page 1: Which properties of soil are you looking to improve using the fly ash? Explain clearly.
Materials and Methods: Did you follow any standard for the experimental testing? This should be mentioned.
What is the reason for adopting water content of 40%? How did you control the temperature and humidity in the curing box?
What was the reason for performing compaction test? Line 63explains that the purpose is to prepare the samples in the next stage. However, looks like it is not the case. Did you compact the samples for UCS testing?
Did you repeat the tests with fly ash and curing agent? If not why?
Table 1: How did you determine the properties of the soil? You need to explain in which condition the shear strength parameters are defined.
If you clearly explain the methodology, figure 1 is unnecessary and can be removed.
Compression and consolidation tests are not the same. Please clarify this confusion in the paper. Also, clearly explain how this test is undertaken. Did you saturate the sample? Did you allow the samples to be drained? How much was the loading rate? Did you follow any standard?
What is meant by: “consolidation stability”?
In the Discussion section, it is required to compare the results with similar studies that used Fly ash and same additives to improve soil properties. Some recent related works are mentioned below. Please compare, elaborate and explain.
Application of Slag–Cement and Fly Ash for Strength Development in Cemented Paste Backfills. Minerals 2019, 9, 22.
Effect of fly ash on the index properties of black cotton soil. Soils Found. 1996, 36, 97–103.
A sulphonated oil for stabilisation of expansive soils. Int. J. Pavement Eng. 2017.
Ground granulated blast furnace slag amended fly ash as an expansive soil stabilizer. Soils Found. 2016, 56, 205–212.
Round 2
Reviewer 1 Report
My comments and suggestions have been fulfilled. Then, I recommend to accept the manuscript for publication and I congratulate the authors for the good job done.
Author Response
Thank you very much for agreeing to receive our manuscript.
Reviewer 2 Report
The paper in its present version is more clear.
I would suggest to replace equations of curing agent, fly ash and water percentage calculations with "by weight of dry soil" next to each mentioned percentage.
The ettringite is not stable at your drying temperature of 110degC - without elemental composition measurements it is difficult to be convinced that your photomicrographs show ettringite. For your future work you may want to consider drying your samples at lower temperature.
Claims that fly ash acts as a curing agent and the concentration of the curing agent may be decreased by the addition of fly ash must be proved by experimental results with LOWER curing agent concentration. The reactivity of fly ash depends on the amount of portlandite released by the curing agent. If you do not have enough portlandite (curing agent) the fly ash may remain non-reacted.
Finally, the table with the composition of your "formulated" curing agent says "cement" - it should be corrected. The text would benefit from another revision to avoid repetitions and correct some grammar.
Reviewer 3 Report
The following revisions are required:
40% looks to be very high as an optimum water content that is obtained in a compaction test. In general, the OWC should be lower than the soil’s PL. It is hard to accept these results. The results of the compaction test should be provided. Can you provide information on another soil with a similar OMC (i.e. 40%) LL and PL with a reference? To help to make the above comment clear, it is required to provide the grain size distribution of the soil (a graph) and then soil classification. You need to mention the standard that is followed. With this, please explain why the soil has very low shear strength properties. I noticed that there are some mistakes in the reference list. For example, in REFERENCE 21, given names are given instead of surnames. Please check the reference list.Author Response
Please see the attachment.
